# Prognostic Factors for Mortality, Activity of Daily Living, and Quality of Life in Taiwanese Older Patients within 1 Year Following Hip Fracture Surgery

**DOI:** 10.3390/jpm12010102

**Published:** 2022-01-13

**Authors:** Ming-Hsiu Chiang, Yu-Yun Huang, Yi-Jie Kuo, Shu-Wei Huang, Yeu-Chai Jang, Fu-Ling Chu, Yu-Pin Chen

**Affiliations:** 1Department of General Medicine, Kaohsiung Chang Gung Memorial Hospital, Kaohsiung 833, Taiwan; b101103050@tmu.edu.tw; 2Department of Nursing & Graduate Institute of Nursing, Chang Gung University of Science and Technology, Kweishan, Taoyuan 333, Taiwan; cutefloat@hotmail.com; 3Department of Orthopedic Surgery, Wan Fang Hospital, Taipei Medical University, Taipei 116, Taiwan; benkuo5@gmail.com (Y.-J.K.); judyya1022@gmail.com (S.-W.H.); 4Department of Orthopedic Surgery, School of Medicine, College of Medicine, Taipei Medical University, Taipei 110, Taiwan; 5Department of Obstetrics and Gynecology, Wan Fang Hospital, Taipei Medical University, Taipei 116, Taiwan; Chvicky1107@gmail.com

**Keywords:** prognostic factors, hip fracture, mortality, activity of daily living, quality of life, older adult assessment

## Abstract

Background. Hip fractures among older adults are a major public health concern worldwide. This study investigated the potential clinical factors that predict postoperative 1-year activities of daily living (ADL), quality of life (QoL), and mortality in Taiwanese older adults following hip fracture. Methods. This is a prospective cohort study enrolling older adults (≥60 years) who had undergone hip fracture surgery in a single medical center. The comprehensive clinical history of each patient was examined. QoL, ADL, and mortality events were recorded consecutively at 3, 6, and 12 months after operation. The multiple logistic regression model and the generalized estimating equation (GEE) were adopted to identify contributing factors for mortality and postoperative ADL and QoL prognosis, respectively. Results. Among 377 participants with hip fracture, 48 died within 1 year of the index operation. ADL and QoL considerably decreased at 3 months following hip surgery. Old age, high Charlson Comorbidity Index, and American Society of Anesthesiologists grading were crucial predictors for mortality at the 1-year follow-up. The generalized estimating equation analysis indicated that the length of postoperative follow-up time, serum albumin level, patient cognitive status, and handgrip strength were considerably associated with QoL and ADL recovery prognosis in the Taiwanese older adults following hip fracture. Conclusions. Hip fractures have long-lasting effects on the older adults. Our data imply several prognosis predicting parameters that may assist clinicians in accounting for an individual’s personalized risks in order to improve functional outcomes and reduce mortality.

## 1. Introduction

Hip fracture is a debilitating disease among older adults [1] and its incidence is increasing. In Taiwan, the incidence of all types of hip fractures increased by 8.6% between 2000 and 2010. In 2018, Taiwan officially became an aged society, and the estimated total number of hip fractures in Taiwan was projected to increase from 18,338 in 2010 to 50,421 by 2035 [2]. Direct costs of hip fracture treatment are enormous, and its subsequent negative effects, including disability, higher risk of cardiovascular disease, and depression, impose a great burden on the patient’s family and are of considerable public health concern [3].

The prognosis of older adults after hip fractures is poor. Activities of daily living (ADLs) and quality of life (QoL) deteriorate significantly after 6 months [4], and the mortality rate is approximately 5.5% at 3 months and 9.1% at 6 months [4] and can be as high as 36% at 1 year postoperatively [5]. Coronary heart disease, pneumonia, and urinary tract infection are major risk factors for mortality in older adults with hip fracture [6]. In terms of functional recovery, only approximately one-fifth of older adults with hip fractures recover their preinjury functional status 1 year postoperatively [4,7]. Therefore, it is critical to identify hip fracture risk factors and organize public health programs to prevent hip fractures.

In patients with hip fracture, postoperative functional recovery and QoL are highly associated with multiple variables, including age, sex, medical complications, baseline Charlson Comorbidity Index (CCI), and sarcopenia [8,9,10]. The influence of surgical delay on postoperative prognosis was deleterious. Mortality significantly increased for patients receiving hip repair surgery >24 hours after admission compared with those operated on within 24 hours [11]. A surgical delay of >21 days had a significantly poorer hip function and QoL than an operation within 7 days of the fracture [12]. Moreover, delays in surgery are correlated with a longer length of hospitalization and a lower possibility of returning to independent living [13]. Handgrip strength is a critical indicator of muscle strength and short-term functional outcome prediction [14]. Predictors of postoperative mortality in patients with hip fracture include age, sex, underlying comorbidities, high American Society of Anesthesiologists (ASA) grading, and prefracture mobility [15]. Patients with poor renal function at presentation are also at an increased risk of death and other complications, including renal, pulmonary, and thromboembolic diseases [16].

With a knowledge of prognostic factors, clinicians can adopt a stratified care approach by prioritizing older adults with hip fractures at a high risk of poor outcomes or high mortality for intensive care [17]. However, considerable variations have been observed among studies in prognostic factors for outcomes following hip fracture surgery in older adults. In addition, most studies have only calculated the changes in ADL or QoL between the preoperative level and that at the last follow-up and may thus have neglected other potential predicting variables if the analysis was conducted in a consecutive manner. This prospective study recorded preinjury and postoperative ADL and QoL at 3, 6, and 12 months and identified the independent prognostic factors of mortality and changes in ADL and QoL in patients with hip fractures aged ≥60 years.

## 2. Methods

### 2.1. Study Design

This prospective cohort study recruited older adults who underwent surgery for hip fracture at a single medical center in Taiwan from 1 January 2017 to 31 December 2019. Qualifying patients were men and women aged ≥60 years who had hip fracture, including intracapsular femoral neck fracture and extracapsular (i.e., basal neck, intertrochanteric, or subtrochanteric) fracture. Patients were excluded if they underwent hip surgery because of a condition other than primary hip fracture, such as osteoarthritis, trauma, tumor metastasis, infection, or avascular necrosis of the femoral head. The study complied with the code of ethics of the World Medical Association (Declaration of Helsinki) and was approved by the Ethics Committee of Taipei Medical University (TMU-JIRB N201709053). Informed consent was obtained from all the participants for publication and participation in the study. 

### 2.2. Measurement of Handgrip Strength

Isometric grip strength, which is the maximum handgrip strength, was measured using a Jamar hydraulic dynamometer (Sammons Preston, Bolingbrook, IL, USA) while the patient was sitting in the bed or on a chair, with the elbow flexed and the wrist in the neutral position, with verbal encouragement. Patients were instructed to squeeze the device three times in each hand as hard as possible [18]. The same investigator conducted measurements for all participants and was blind to the clinical data. For each participant, the best of six measurements was used. A handgrip strength of <26 kg for men and <18 kg for women is considered low, based on the threshold values recommended by the Asian Working Group for Sarcopenia [19].

### 2.3. Measurement of Other Clinical Parameters

Basic demographic data, including age, sex, and body mass index, were collected from medical records for analysis, along with preoperative laboratory data, including hemoglobin level, serum creatinine, sodium, and albumin levels. The underlying comorbidities were presented as CCI. We also recorded the specific amount of time elapsed between a patient’s fall and the time of operation. The operation records of each patient were screened to extract the following information: ASA grade, type of hip fracture repair surgery, and blood loss during the operation. Bone mineral density (BMD) T-score and handgrip strength were obtained within 1 week after surgery (i.e., T-score obtained through dual-energy X-ray absorptiometry (DXA)) [10]. After patient consent was obtained, interviews were conducted with the patients and their caregivers on the patients’ admission for hip fracture surgery by using the Short Portable Mental Status Questionnaire (SPMSQ) [20] for screening dementia. For assessing preinjury and postoperative (at 3, 6, and 12 months) performance in ADL and QoL, the Barthel Index (BI) and EuroQol-5D questionnaire (EQ-5D) [21] were adopted, respectively [22]. The BI has an ordinal scale with scores from 0 to 100 and is calculated using 10 variables that represent ADL and mobility [22]. A higher value is associated with a greater likelihood of independent community living. The Chinese version of the EQ5D was used in this study and it exhibited a high level of agreement (intraclass correlation coefficients >0.75) and convergent validity (Pearson’s correlation coefficients >0.95) with the value sets of the versions of the EQ-5D from the UK, Japan, and Korea [23]. The Chinese version of the BI has been validated with moderate to excellent agreement among raters for individual items (kappa: 0.53–0.94) and total score (intraclass correlation coefficient Z: 0.94) [24]. 

## 3. Statistical Analysis

All statistical analyses were conducted using SPSS Statistics for Windows, version 18.0 (SPSS, Chicago, IL, USA). Categorical variables are presented as frequencies and percentages and continuous variables are presented as mean ± standard deviation. Univariate analyses were conducted on numerous potential risk factors for mortality between survival and mortality subgroups 1 year after hip fracture surgery; a chi-squared test was used to compare categorical variables, whereas an independent t-test was used to compare continuous variables. Factors that differed significantly (*p* < 0.05) in the univariate analysis were included in the multiple logistic regression model to identify critical mortality-contributing factors. In terms of comparing repeated ADL measurements at postoperative 3, 6, and 12 months with preinjury status, a paired *t*-test was used. Postoperative EQ-5D and BI were measured repetitively, so a generalized estimating equation (GEE) was adopted to screen all potential influencing clinical parameters. For cases that were lost to follow-up, missing data imputation was performed. For all tests, two-sided *p* < 0.05 was considered statistically significant.

## 4. Results

A total of 416 older adults were screened; two were suspected of pathologic fracture, four of femoral head avascular necrosis, 15 were less than 60 years old, and 18 refused to participate. At last, 377 older adults with hip fracture, comprising 105 men (28%) and 272 women (72%), were enrolled. Appendix A shows the study flowchart and number of patients staying in contact at each follow-up. Overall, 56 participants lost contact after 1 year of follow-up, resulting in a 1-year loss to follow-up rate of around 14.8%. The mean age was 81 ± 9.5 years (range: 60–103 years). The average body mass index was 22 ± 3.6, and the mean CCI was 4.8 ± 1.8. Of the 377 patients, 203 had femoral neck fracture (54%) and 174 had intertrochanteric fracture (46%). Upon admission, the patients had a mean handgrip strength of 12 ± 9.4 kg. The average ASA grade was 2.7 ± 0.5, and the mean surgical delay from the index fracture was 77 ± 234 h. Furthermore, 236 patients (63%) received open reduction and internal fixation, and the remaining underwent joint replacement surgery. Table 1 summarizes the patients’ baseline clinicodemographic characteristics.

In all, 48 patients died within 1 year after surgery; thus, the 1-year mortality rate after hip fracture was approximately 12.7%. Univariate analysis revealed that the following clinical variables were significantly correlated with postoperative 1-year mortality: age, CCI, preoperative hemoglobin value, serum albumin level, SPMSQ score, handgrip strength, and ASA grade (Appendix A). However, only age, CCI, and ASA grade were significant in the multiple logistic regression model (Table 2).

A significant decrease in QoL and ADL was observed at postoperative 3, 6, and 12 months compared with the patients’ preinjury status. The preinjury EQ-5D score was 0.84 and decreased to 0.73, 0.76, and 0.77 at postoperative 3, 6, and 12 months, respectively (all *p* < 0.001; Figure 1), and BI decreased from 84.4 to 68.97, 71.27, and 71.11, respectively, without signs of recovery to their preinjury status (all *p* < 0.001; Figure 2). We assessed the impact of different clinical parameters on postoperative EQ-5D scores and BI by using GEE. In the univariate analysis, age, CCI, BMD T-score, preoperative hemoglobin value, serum albumin level, SPMSQ score, ASA grade, length of postoperative follow-up time, and handgrip strength were independently associated with postoperative EQ-5D score and BI (Appendix A). However, multivariate analysis with multiple imputations revealed that only the serum albumin level, SPMSQ score, handgrip strength, and length of postoperative follow-up time had a significant impact on both postoperative EQ-5D scores and BI (Table 3). Higher serum albumin level and handgrip strength were associated with better improvements in QoL and ADL, whereas higher SPMSQ score was inversely related to poorer outcomes. In terms of postoperative follow-up time, a significant reduction in QoL and ADL persisted up to postoperative 1 year compared with the patients’ preinjury status.

## 5. Discussion

Several key epidemiological observations were noted in the present study. The overall participant’s mean age was 81 years old, with female predominance. Similarly, one epidemiological study from Taiwan indicated that the average age of hip fracture was 76.7 ± 9.0 years for women and 74.1 ± 9.6 years for men [25]. Anemia, malnutrition, and osteoporosis were prevalent in all participants. Moreover, the average SPMSQ score was 3.5, which indicated that over half of the total number of participants had mild dementia. Finally, the mean handgrip strength was 9.4 kg in the present study, which was much lower than the sarcopenia diagnostic criteria (<26 kg in men and <18 kg in women), implying a high incidence of concomitant sarcopenia and hip fracture. Consistently, a recent cross-sectional study reported that over half of older adults with hip fracture had sarcopenia [10]. In other words, older adults with hip fracture are more fragile and may thus require more healthcare resources.

Apart from old age [26], various clinical variables have been regarded as risk factors for mortality after hip fracture repair surgery. Li et al. concluded that the odds of postoperative 1-year mortality was 3.35 in patients whose CCI scores were >3 [27]. In addition, patients with a CCI score of ≥3 sustained an increased 19% risk of recurrent osteoporotic fracture compared with those with a CCI score of 0 at 1 year after the hip fracture [28]. Lower handgrip strength has also been associated with higher postoperative mortality [29]. However, in our study, handgrip strength was a significant predictor for postoperative mortality only in the univariate analysis and not in the multiple logistic regression model. This may be because the patients’ overall handgrip strength was generally low. Furthermore, the high correlation between age, CCI, and ASA grade may be another possible reason (R = −0.28, −0.27, and −0.24, respectively; all *p* < 0.01) that overshadowed the predictability of handgrip strength. More well-organized prospective studies are warranted to determine the reliability of handgrip strength in predicting mortality after hip fracture repair surgery.

Hip fracture can considerably impair the QoL and mobility of older adults. Hershkovitz et al. reported that <10% of patients with hip fracture are functionally independent after 1 year of postoperative rehabilitation [29]. Health-related QoL also severely deteriorates after hip fracture. Using the EQ-5D, Amarilla-Donoso et al. reported a significant reduction in all dimensions from prefracture status to postoperative 1 month [30]. In the present study, after 1 year of follow-up, we observed that both EQ-5D scores and BI reached the lowest level at postoperative 3 months. Although QoL and ADL later improved in most participants, they remained significantly lower than their preinjury status.

Multiple clinical parameters have been proposed as predictors for postoperative ADL recovery, including age, sex, surgical delay, preinjury status, [31,32,33] preoperative hemoglobin values, renal function, and handgrip strength [34,35]. In the present study, nine clinical variables were identified to be independently associated with postoperative BI in the GEE univariate analysis. However, only the length of postoperative follow-up time, serum albumin level, SPMSQ score, and handgrip strength were found to be significant in the GEE multivariate analysis model with multiple imputation. Dementia, delirium, and malnutrition are also critical risk factors for poor postoperative outcomes following hip fracture [36,37]. Handgrip strength is assessed noninvasively and is thus convenient for repetitive measurements. It is not only a part of sarcopenia diagnostic criteria but has also been drawing attention as a prognostic indicator for postoperative ADL in recent years. Choi et al. enrolled 242 older adults with hip fracture and recorded their respective handgrip strength with a hand dynamometer. They reported that handgrip strength is predictive of postoperative complications and can therefore be useful for preoperative screening [38]. Di Monaco et al. also reported that handgrip strength assessed before rehabilitation can independently predict functional outcomes after a 6-month follow-up in women with hip fracture [39]. In summary, the role of handgrip strength as a prognostic factor after hip fracture in older adults is gaining attention. Our findings corroborate that handgrip strength is significantly correlated with postoperative ADL changes. 

The role of sex in predicting postoperative ambulatory prognosis was heterogeneous. Some studies concluded sex is a major contributing factor [8,32]. Findings in the present study and others reported that man and woman had similar functional recovery or ability to return back to community lives [40,41]. We surmised that this discrepancy may be resulting from distinct population characteristics included in each study. For example, the mean age and number of comorbidities differed significantly between male and female, which could impact the potential for recovery. In the present study there were no significant differences in age or CCI between male and female population. Long surgical delay after a falling accident was an important predictor for mortality and morbidity risk. However, in the present study, delay in surgery from fracture was not found to be correlated with mortality or ADL and QoL prognosis. One possible underlying etiology was that nearly 80% of the total participants were quickly sent to our hospital after injury and they received operation within 72 hours following hip fracture; only a minority of patients had prolonged surgical delay. BMD T-score was only identified as significant clinical parameters in the univariate analysis. One possible reason was the instant osteoporosis treatment prescription once osteoporosis was diagnosed. A recent study has reported that osteoporosis therapy incurred significant benefits on the patient’s functional recovery [41]. As a result, the heterogeneous impact of lower BMD T-score may be hindered by the prompt initiation of osteoporosis treatment.

## 6. Limitations

This study has several limitations. First, only 377 participants were enrolled, and they may not represent all the older adults with hip fracture in Taiwan. Second, loss to follow-up is inevitable and when loss to follow-up exceeds 20%, it may pose serious threats to the study validity [42]. In the present study, 1-year loss to follow-up rate was around 14.8% and missing data imputation was performed in GEE for compensation. However, the results should still be interpreted cautiously. Third, this study was conducted in a metropolitan area, and the findings may be region-specific and may not represent the general epidemiology of hip fracture in Taiwan. Finally, data were only collected before hip fracture surgery and within 1 year after surgery. Long-term follow-up studies are warranted to follow the natural course of hip fractures and their impact on QoL and motility in this specific cohort.

## 7. Conclusions and Implications

By prospectively recruiting older adults who underwent surgery for hip fracture and following the postoperative changes in ADL and QoL, this study identified potential predicting clinical parameters. The postoperative 1-year mortality rate was approximately 12.7% in this study; both BI and EQ-5D declined severely at postoperative 3 months and recovered gradually afterwards. Age, CCI, and ASA grade were good predictors of postoperative mortality within 1 year. In addition, the length of postoperative follow-up time, serum albumin level, SPMSQ score, and handgrip strength were independent factors correlated with postoperative QoL and ADL; older adults continuously had poorer functional outcomes compared with their preinjury status up to 1 year after receiving hip fracture operation. Higher serum albumin level, hand grip strength and lower SPMSQ score were correlated with better postoperative QoL and ADL. Our findings may guide the establishment of a robust prediction and screening program that could improve functional outcomes and reduce the risk of mortality in older adults with hip fracture.

## Figures and Tables

**Figure 1 jpm-12-00102-f001:**
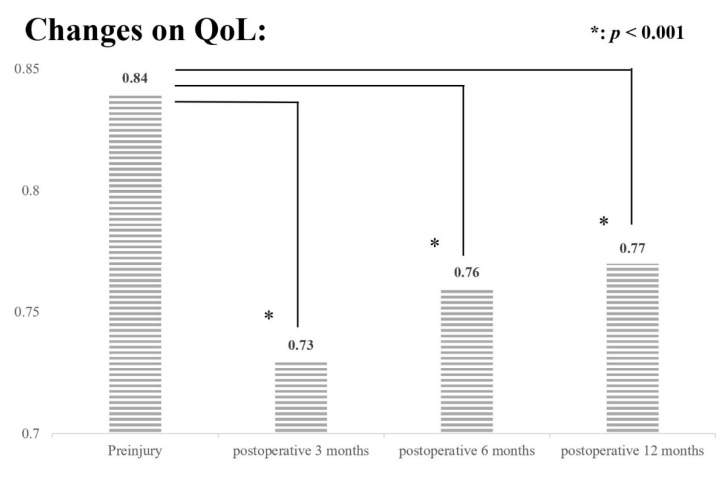
Consecutive QoL (EQ-5D) measurements at preinjury, 3, 6, and 12 months after hip fracture repair surgery. *: *p* < 0.001.

**Figure 2 jpm-12-00102-f002:**
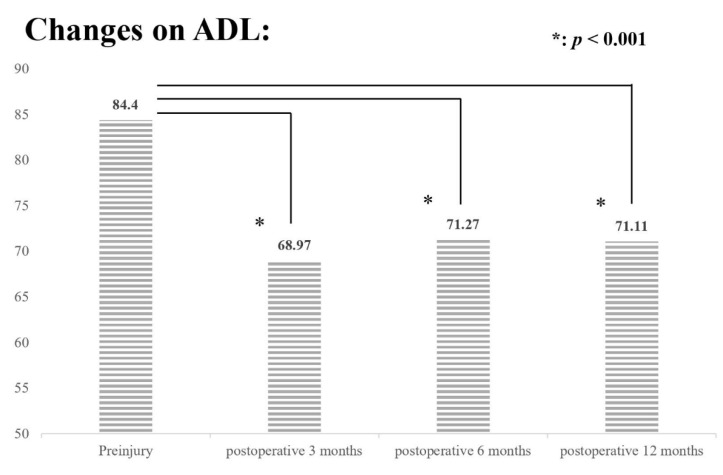
Consecutive ADL (BI) measurements at preinjury, 3, 6, and 12 months after hip fracture repair surgery. *: *p* < 0.001.

**Table 1 jpm-12-00102-t001:** Baseline clinicodemographic characteristics of the study population.

Clinical Characteristics (*n* = 377)	Mean ± SD/Number (Percentage)
**Age**	81 ± 9.5
**Gender**	
Male	105 (28%)
Female	272 (72%)
**BMI**	22 ± 3.6
**CCI**	4.8 ± 1.8
**Types of fracture**	
Femoral neck fracture	203 (54%)
Pertrochanteric fracture	174 (46%)
**ASA grading**	2.7 ± 0.52
**SPMSQ score**	3.5 ± 3.7
**Handgrip strength (kg)**	12 ± 9.4
**BMD T-score**	−3.8 ± 1.1
**Pre-operation laboratory data**	
Hb (g/dL)	12 ± 2.3
Na (mmol/L)	137 ± 4.1
Creatinine (mg/dL)	1.2 ± 1.2
Vitamin D3 (ng/mL)	20 ± 16
(*n* = 260)	
Parathyroid hormone (pg/mL)	55 ± 62
(*n* = 308)	
Albumin (g/dL)	3.1 ± 0.4
(*n* = 336)	
**Surgical record**	
Surgical delay (hour)	77 ± 234
265 patients (70%) ≤ 48 h	
294 patients (78%) ≤ 72 h	
Operation time (min)	79 ± 41
Blood loss (cc)	111 ± 109
**Surgical classification**	
ORIF	236 (63%)
Joint replacement	141 (37%)

Abbreviations: ASA, American Society of Anesthesiologists; BMI, body mass index; BMD, bone mineral density; CCI, Charlson Comorbidity Index; Hb, hemoglobin; ORIF, open reduction and internal fixation; SD, standard deviation; SPMSQ, Short Portable Mental Status Questionnaire.

**Table 2 jpm-12-00102-t002:** Multiple logistic regression model of factors affecting postoperative 1-year mortality in older adults with hip fracture.

Variables	Β	*p*-Value	Odds Ratio	95% CI Odds Ratio	
				Lower	Upper
**Age**	**0.07**	**0.016**	**1.07**	**1.01**	**1.1**
**CCI**	**0.22**	**0.03**	**1.3**	**1.02**	**1.5**
Pre-operation Hb (g/dL)	−0.16	0.11	0.85	0.70	1.04
Albumin (g/dL)	−0.22	0.7	0.81	0.30	2.2
SPMSQ score	0.09	0.09	1.1	0.97	1.2
Handgrip strength (kg)	−0.04	0.3	0.96	0.89	1.04
**ASA grading**	**1.5**	**0.03**	**4.6**	**1.1**	**18.7**

Abbreviations: ASA, American Society of Anesthesiologists; CCI, Charlson Comorbidity Index; CI, confidence interval; Hb, hemoglobin; SPMSQ, Short Portable Mental Status Questionnaire.

**Table 3 jpm-12-00102-t003:** GEE model with multiple imputation of factors affecting postoperative EQ-5D and ADL in older adults with hip fracture.

Variables	β	*p*-Value	95% CI Ratio	
			Lower	Upper
**EQ-5D**				
**Postoperative 3 months vs. preinjury status**	**−0.31**	**<0.001**	**−0.35**	**−0.27**
**Postoperative 6 months vs. preinjury status**	**−0.25**	**<0.001**	**−0.29**	**−0.20**
**Postoperative 12 months vs. preinjury status**	**−0.23**	**<0.001**	**−0.27**	**−0.18**
Age	−0.001	0.34	−0.004	0.001
CCI	−0.008	0.25	−0.021	0.005
BMD T-score	0.007	0.48	−0.012	0.025
Pre-operation Hb (g/dL)	0.005	0.40	−0.006	0.016
**Albumin** (g/dL)	**0.060**	**0.018**	**0.010**	**0.11**
**SPMSQ**	**−0.019**	**<0.001**	**−0.025**	**−0.013**
ASA grading	0.005	0.81	−0.038	0.048
**Handgrip strength (kg)**	**0.003**	**0.01**	**0.001**	**0.005**
**ADL**				
**Postoperative 3 months vs. preinjury status**	**−32.6**	**<0.001**	**−36.8**	**−28.4**
**Postoperative 6 months vs. preinjury status**	**−25.6**	**<0.001**	**−30.3**	**−20.9**
**Postoperative 12 months vs. preinjury status**	**−25.5**	**<0.001**	**−30.2**	**−20.8**
Age	0.11	0.43	−0.16	0.39
CCI	−0.85	0.24	−2.3	0.56
BMD T-score	0.66	0.52	−1.34	2.7
Pre-operation Hb (g/dL)	0.5	0.43	−0.74	1.7
**Albumin** (g/dL)	**8.8**	**0.002**	**3.2**	**14.3**
**SPMSQ**	**−2.9**	**<0.001**	**−3.5**	**−2.26**
ASA grading	−2.1	0.38	−6.9	2.6
**Handgrip strength** (kg)	**0.29**	**0.017**	**0.05**	**0.52**
Hip fracture type (PTF vs. FNF)	−2.5	0.27	−7.0	1.9

Abbreviations: ASA, American Society of Anesthesiologists; BMD, bone mineral density; CCI, Charlson Comorbidity Index; CI, confidence interval; FNF, femoral neck fracture; Hb, hemoglobin; PTF, pertrochanteric fracture; SPMSQ, Short Portable Mental Status Questionnaire.

## Data Availability

Due to the sensitive nature of the questions asked in this study, survey respondents were assured that raw data would remain confidential and would not be shared.

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
