# Peer review of "Prognostic Factors for Mortality, Activity of Daily Living, and Quality of Life in Taiwanese Older Patients within 1 Year Following Hip Fracture Surgery"

_jpm, 2022, doi:10.3390/jpm12010102_

Round 1

Reviewer 1 Report

Dear sirs:

Thank for your work about prognostics factors affecting mortality and functional status after hip fracture surgery. This information could be helpful to improve assistance to these people but I consider your manuscript needs some changes to be improved.

Title: The term “geriatric” is expressed only in title and abstract while the term “older people” is used in your manuscript body. Last sentence of your introduction informs that you refer to people “aged ≥60 years”. I don’t know if this is the age defined as geriatric in Taiwan, but this is not the case in other countries. Please, make your title more explicit including the country name and according to Taiwan geriatric age definition i.e.: “…Quality of life in Taiwan Geriatric Patients…” or “…Quality of life in Taiwan Older Patients…”. This definition must be accordingly adjusted in abstract and, if needed, in the manuscript body.

Abstract, line 21: In addition to changes as per title adjust, add the specific age of your sample “This is a prospective cohort study enrolling older adults (≥60 years) who…”

Methods, lines 84-85: “was approved” is repeated twice.

Methods, lines 105-115: Here, you stated that SPMSQ and EQ-5D were used to asses “preinjury” status, but EQ-5D was also used to asses postoperative status, please correct. For Barthel Index you used a Chinese version. You used a validated Chinese version for EQ-5D questionnaire?

Statistical Analysis: please group correctly the statistics used to assed factors affecting mortality and the ones used to assed factors affecting ADL and QoL. For example, you stated that chi-square test or independent t-test were used but, between which independent groups? For mortality contribution? Between this paragraph and the logistic regression model used to complete the mortality contribution analysis there is another paragraph about ADL analysis.

Statistical Analysis, lines 120-122: You used paired t-test for compare repeated ADL measurements. Why not for repeated QoL measurements? Any case, repeated measures ANOVA is a more adequate statistic.

Results, line 132: study flowchart was not provided.

Results, line 143 Table 1: In this table it appears for the first time “BMD t-score”. I imagine it was measure pre injury and you find this data in patients’ history. Did you collect this data for all your sample population? Please, include this information in the method section (e.g. point 2.3)

Results, lines 144-145 Table 1: You stated some of the abbreviations used, but some other not for example, SPMSQ nor BMD. Also, if you decide to use abbreviations in table body, please be consequent. Charlson Comorbidity Index (CCI) is stated both in abbreviation and complete name. The same commentary will be applied for lines 154-155 Table 2, and lines 173-174 for Table 3.

Results, line 150: supplementary table 1 was not provided.

Results, line 152 Table 2: The word “Age” is in bold font. I imagine that CCI and ASA will be also bold.

Results, line 162: “T-score” I imagine it “BMD T-score”.

Results, line 163: what is “length of postoperative recovery”? Your own follow-up length? See also Discussion section, point 2.

Results, lines 165-165: supplementary tables 2 and 3 were not provided.

Limitations, lines 232-233: You state in this point the % of follow-up loses, but this data must be first included in the results section including the causes if knowing.

Limitations, line 237. Barthel index is “subjective”. Is EuroQol objective? I do not consider a limitation the use of validated instruments to measure the outcomes.

Discussion section.

1) As you stated in your introduction section, sex (maybe “gender” as indicated in table 1, would be more appropriated) was associated with functional recovery and mortality by other authors but, as per your results section, this association was not found in your study. I consider this is an important question to be discussed.

2) Your data show a recovery during your follow up but QoL and ADL remain lower that preinjury values, and I’m not sure if the association seen between several variables is related to the recovery during your follow-up or to the lost of QoL and ADL at one year comparing with preinjury values.

As indicated in your abstract “serum albumin level, patient cognitive status, and handgrip strength were considerably associated with QoL and ADL recovery in the Taiwanese geriatric hip fracture population” but your discussion section compares most with other studies which found poor functional outcomes after hip fracture.

Your Conclusion section state: “In addition, length of postoperative recovery time, serum albumin level, SPMSQ score, and handgrip strength were independent factors correlated with postoperative QoL and ADL.” With postoperative “recovery” during one year or with postoperative lost?

Also, your data show some recovery between 3 and 6 months but less recovery between 6 and 12 months. If you made a repeated measures ANOVA you can inform if the recovery is statistically significant or not, and with-in which follow-up period.

Regards.

Author Response

Submission no: jpm-1496209 

Submission title: Prognostic Factors for Mortality, Activity of Daily Living, and Quality of Life in Taiwan Older Patients Within 1 Year Following Hip Fracture Surgery

Dear editors from journal of Personalized Medicine:

Thank you for giving us the opportunity to revise our submitted manuscript. We tried our best to reply all the reviewer’s comments point-to-point and edit the manuscript accordingly. We hoped that the revised version fulfills the requirements of your esteemed journal.

Please kindly express our sincere and grateful appreciation to the anonymous reviewers. At last, please inform us if additional revisions are needed.

Academic Editor

Question 1:

Good manuscript. Not all methods mentioned in the results are referred to in the methods concerning the abstract. That should be changed.

Reply 1:

Thank you very much for reviewing this study and giving us great suggestions to improve our work. We have added statistical analysis methods into the abstract. (lines 24-26)

Question 2:

What is exactly meant by prospective study in this survey?

Reply 2:

This study was initiated in 2017. We have recruited participants and collected data in a prospective manner by interviewing and reexamining each patient on three specific postoperative timing (perioperative, postoperative 3, 6, and 12 months).

Question 3:

The conclusions should be detailed. What do you mean by personalized approach? 

Reply 3:

We have reedited the conclusion section in abstract. (lines 33-36) We hoped that by identifying potential prognostic factors, we could establish a screening system that rate risk scores for each older adult following a hip fracture and provide stratified care to reduce mortality and improve their functional outcomes.

Reviewer 1

Question 1:

Thank for your work about prognostics factors affecting mortality and functional status after hip fracture surgery. This information could be helpful to improve assistance to these people but I consider your manuscript needs some changes to be improved.

 Title: The term “geriatric” is expressed only in title and abstract while the term “older people” is used in your manuscript body. Last sentence of your introduction informs that you refer to people “aged ≥60 years”. I don’t know if this is the age defined as geriatric in Taiwan, but this is not the case in other countries. Please, make your title more explicit including the country name and according to Taiwan geriatric age definition i.e.: “…Quality of life in Taiwan Geriatric Patients…” or “…Quality of life in Taiwan Older Patients…”. This definition must be accordingly adjusted in abstract and, if needed, in the manuscript body.

Reply 1:               

Thank you very much for giving us so much constructive advice and precious recommendations. We have changed our title and changed the term “geriatric patients” to “older adults” throughout the abstract and manuscript.

Question 2:

Abstract, line 21: In addition to changes as per title adjust, add the specific age of your sample “This is a prospective cohort study enrolling older adults (≥60 years) who…”

Reply 2:

Thank you for the suggestion. We have added the specific age into abstract. (line 21)

Question 3:

Methods, lines 84-85: “was approved” is repeated twice.

Reply 3:

We are sorry for the typing error. The duplicate words were removed. (line 91)

Question 4:

Methods, lines 105-115: Here, you stated that SPMSQ and EQ-5D were used to asses “preinjury” status, but EQ-5D was also used to asses postoperative status, please correct. For Barthel Index you used a Chinese version. You used a validated Chinese version for EQ-5D questionnaire?

Reply 4:

Thank you for the advice. We have corrected the statement. In addition, we added a statement regarding the validity of the Chinese version for EQ-5D used in this study (lines 117-119).

Question 5:

Statistical Analysis: please group correctly the statistics used to assed factors affecting mortality and the ones used to assed factors affecting ADL and QoL. For example, you stated that chi-square test or independent t-test were used but, between which independent groups? For mortality contribution? Between this paragraph and the logistic regression model used to complete the mortality contribution analysis there is another paragraph about ADL analysis.

Reply 5:

We are sorry for the ambiguity found in the statistical analysis paragraph. The chi-square test and independent t-test were used in univariate analysis for identifying potential mortality contributing factors. Paired t-test was used to compare postoperative ADL measurements at postoperative 3, 6, and 12 months with preinjury status. At last, generalized estimating equation (GEE) was adopted to screen all potential clinical parameters influencing ADL and QoL changes (lines 134-140).

Question 6:

Statistical Analysis, lines 120-122: You used paired t-test for compare repeated ADL measurements. Why not for repeated QoL measurements? Any case, repeated measures ANOVA is a more adequate statistic.

Reply 6:

We have considered using ANOVA tests initially. However, loss of contacts occurred at each postoperative follow up period, inevitably resulting in missing data. ANOVA tests could not perform missing data imputation and would thus lose much statistical power.

Question 7:

Results, line 132: study flowchart was not provided.

Reply 7:

We are sorry for the mistake. We have provided the supplementary material in the revised manuscript.

Question 8:

Results, line 143 Table 1: In this table it appears for the first time “BMD t-score”. I imagine it was measure pre injury and you find this data in patients’ history. Did you collect this data for all your sample population? Please, include this information in the method section (e.g. point 2.3)

Reply 8:

BMD T-score and isometric grip strength were collected within one week after operation. We collected BMD T-score using DXA for all participants. (lines 111-113)

Question 9:

Results, lines 144-145 Table 1: You stated some of the abbreviations used, but some other not for example, SPMSQ nor BMD. Also, if you decide to use abbreviations in table body, please be consequent. Charlson Comorbidity Index (CCI) is stated both in abbreviation and complete name. The same commentary will be applied for lines 154-155 Table 2, and lines 173-174 for Table 3.

Reply 9:

We have corrected all abbreviations used in the three tables.

Question 10:

Results, line 150: supplementary table 1 was not provided.

Reply 10:

We are sorry for the mistake. We have provided supplementary table 1 in the revised manuscript.

Question 11:

Results, line 152 Table 2: The word “Age” is in bold font. I imagine that CCI and ASA will be also bold.

Reply 11:

The mistake was corrected. Thank you very much.

Question 12:

Results, line 162: “T-score” I imagine it “BMD T-score”.

Reply 12:

The mistake was corrected. (line 182)

Question 13:

Results, line 163: what is “length of postoperative recovery”? Your own follow-up length? See also Discussion section, point 2.

Reply 13:

We are sorry for the incorrect wording. We have replaced the word into “length of postoperative follow-up time” throughout the whole manuscript, which would be more appropriate

Question 14:

Results, lines 165-165: supplementary tables 2 and 3 were not provided.

Reply 14:

We are sorry for the mistake. We have provided supplementary tables 2 and 3 in the revised manuscript.

Question 15:

Limitations, lines 232-233: You state in this point the % of follow-up loses, but this data must be first included in the results section including the causes if knowing.

Reply 15:

We have added % of loss of follow-up in the Result section. (lines 150-151) Moreover, we have rechecked participant numbers and found out that the loss of follow-up percentage should be 14.8% instead of 14.6%. Corrections were made throughout the manuscript.

Question 16:

Limitations, line 237. Barthel index is “subjective”. Is EuroQol objective? I do not consider a limitation the use of validated instruments to measure the outcomes.

Reply 16:

After discussion, all authors agreed to remove these sentences in the Limitation section.

Question 17:

1) As you stated in your introduction section, sex (maybe “gender” as indicated in table 1, would be more appropriated) was associated with functional recovery and mortality by other authors but, as per your results section, this association was not found in your study. I consider this is an important question to be discussed.

Reply 17:

We have added new paragraph in the Discussion section to discuss several important variables (sex, surgical delay, and osteoporosis) that may be associated with clinical prognosis following hip fracture surgery. (lines 253-271)

Question 18:

2) Your data show a recovery during your follow up but QoL and ADL remain lower that preinjury values, and I’m not sure if the association seen between several variables is related to the recovery during your follow-up or to the lost of QoL and ADL at one year comparing with preinjury values.

As indicated in your abstract “serum albumin level, patient cognitive status, and handgrip strength were considerably associated with QoL and ADL recovery in the Taiwanese geriatric hip fracture population” but your discussion section compares most with other studies which found poor functional outcomes after hip fracture.

Your Conclusion section state: “In addition, length of postoperative recovery time, serum albumin level, SPMSQ score, and handgrip strength were independent factors correlated with postoperative QoL and ADL.” With postoperative “recovery” during one year or with postoperative lost?

Also, your data show some recovery between 3 and 6 months but less recovery between 6 and 12 months. If you made a repeated measures ANOVA you can inform if the recovery is statistically significant or not, and with-in which follow-up period.

Reply 18:

We have added more detailed interpretation for the results of the GEE model identifying factors affecting postoperative EQ-5D and BI in the Result section (Table 3). (lines 188-192) Serum albumin level and hand grip strength were positively correlated with postoperative QoL and ADL whereas SPMSQ score was inversely associated. No matter shorter (3 months) or longer postoperative follow-up time (6 and 12 months, compared to preinjury status, significantly decreased QoL and ADL were observed in these three time points.

We have edited the abstracts to imply that serum albumin level, patient cognitive status, and handgrip strength were considerably associated with QoL and ADL “recovery prognosis”. Malnutrition, dementia, and poor cognitive condition were identified in other studies to be related with poorer QoL and ADL after hip fracture surgery, which were in accordance with the findings of the present study. Similarly, higher hand grip strength is positively correlated with better functional outcomes.

ANOVA test was not suitable in the present study due to lack of intact patient’s data and it would be lacking statistical power.

Reviewer 2 Report

I would like to thank the authors for allowing me to review their manuscript. The authors have prospectively collected quality of life scores, an outcome measure that is often lacking in the hip fracture literature. As registries move towards collecting QOL indicators, we may know more about treatment modalities and their influence on outcomes, however, until this point, studies such as this have their role in directing policy, particularly in advanced countries such as Taiwan that are becoming aged. Seeing graphical illustration of QOL such as in figure 1 highlights the need for improving care, but also setting family expectations. Annual mortality is low, which is consistent with other Asian nations. The mean age of fracture is consistent with international literature.

I don’t have any major issues with the article. The regression analysis is fed by simple t tests. CCI & ASA could be portrayed as a categorical and not continuous in the regression analysis, but the message is that the sicker the patient, the more chance of dying. The use if imputation for a admirable loss to follow up of under 15% is appropriate.

Overall, I think this study is useful for the Taiwanese population particularly and can be extrapolated to other Asian nations.

Author Response

Submission no: jpm-1496209 

Submission title: Prognostic Factors for Mortality, Activity of Daily Living, and Quality of Life in Taiwan Older Patients Within 1 Year Following Hip Fracture Surgery

Dear editors from journal of Personalized Medicine:

Thank you for giving us the opportunity to revise our submitted manuscript. We tried our best to reply all the reviewer’s comments point-to-point and edit the manuscript accordingly. We hoped that the revised version fulfills the requirements of your esteemed journal.

Please kindly express our sincere and grateful appreciation to the anonymous reviewers. At last, please inform us if additional revisions are needed.

Reviewer 2

Question 1:

I would like to thank the authors for allowing me to review their manuscript. The authors have prospectively collected quality of life scores, an outcome measure that is often lacking in the hip fracture literature. As registries move towards collecting QOL indicators, we may know more about treatment modalities and their influence on outcomes, however, until this point, studies such as this have their role in directing policy, particularly in advanced countries such as Taiwan that are becoming aged. Seeing graphical illustration of QOL such as in figure 1 highlights the need for improving care, but also setting family expectations. Annual mortality is low, which is consistent with other Asian nations. The mean age of fracture is consistent with international literature.

I don’t have any major issues with the article. The regression analysis is fed by simple t tests. CCI & ASA could be portrayed as a categorical and not continuous in the regression analysis, but the message is that the sicker the patient, the more chance of dying. The use if imputation for a admirable loss to follow up of under 15% is appropriate.

Overall, I think this study is useful for the Taiwanese population particularly and can be extrapolated to other Asian nations.

Reply 1:               

Thank you for reviewing this study and giving us great advice and feedbacks. We hoped that results of the current study could aid in the establishment of hip fracture prevention and caring program in the future.

Reviewer 3 Report

Review for Journal of Personalized Medicine

Title: Prognostic Factors for Mortality, Activity of Daily Living, and 2 Quality of Life in Geriatric Patients Within 1 Year Following 3 Hip Fracture Surgery

Summary:

The authors present an analysis of ADL and QoL after hip fractures in a prospective cohort study of a geriatric population. The topic is of great interest for our daily work, because of the mentioned consequences for these patients and the economic burden. The authors described prognostic factors for mortality and changes of ADL and QoL in their patient cohort in a single centre.

General notes:

  • Introduction: The authors mentioned a significant difference in mortality of delayed surgery after more than 21 days. The increase of mortality after 24h should also be mentioned in my opinion.
  • Results: The mean surgical delay was 77h in this study. This delay is difficult to accept in my opinion. There are many studies, that show an increased mortality of patient with surgery >24h after admission (e. g. Welford et al, Bone Joint J 2021 Jul;103-B(7):1176-1186. doi: 10.1302/0301-620X.103B7.BJJ-2020-2582.R1). Can you explain the delay of 77h in your study? Also 63% were treated with ORIF (only 46% were extracapsular fractures), where the time to surgery plays an important role for postoperative complications like avascular necrosis of the femoral head in case of intracapsular fractures.
  • Discussion: Not all important variables that are mentioned in this study are discussed in the manuscript (e. g. surgical delay, osteoporosis).

Specific notes:

  • What’s your definition of “length of postoperative recovery”? Length of hospital stay?

Summarised assessment:

The present manuscript covers an interesting and relevant topic in trauma surgery. There are no major drawbacks in the methodology of the study. The study is well written. My only concern relates to the surgical delay in this study. A surgical delay of several days is an important factor for reduced outcome of these patients in my opinion. This point is not discussed in the manuscript. Without an explanation of this fact and an analysis of its role in the interpretation of this study, I can’t recommend to accept this paper for publication.

Author Response

Submission no: jpm-1496209 

Submission title: Prognostic Factors for Mortality, Activity of Daily Living, and Quality of Life in Taiwan Older Patients Within 1 Year Following Hip Fracture Surgery

Dear editors from journal of Personalized Medicine:

Thank you for giving us the opportunity to revise our submitted manuscript. We tried our best to reply all the reviewer’s comments point-to-point and edit the manuscript accordingly. We hoped that the revised version fulfills the requirements of your esteemed journal.

Please kindly express our sincere and grateful appreciation to the anonymous reviewers. At last, please inform us if additional revisions are needed.

Reviewer 3

Question 1:

Introduction: The authors mentioned a significant difference in mortality of delayed surgery after more than 21 days. The increase of mortality after 24h should also be mentioned in my opinion.

Reply 1:               

Thank you for reviewing this study and giving us many precious suggestions for improvement. We have mentioned the profound impact of surgical delay on mortality in the Introduction section. (lines 59-61)

Question 2:

Results: The mean surgical delay was 77h in this study. This delay is difficult to accept in my opinion. There are many studies, that show an increased mortality of patient with surgery >24h after admission (e. g. Welford et al, Bone Joint J 2021 Jul;103-B(7):1176-1186. doi: 10.1302/0301-620X.103B7.BJJ-2020-2582.R1). Can you explain the delay of 77h in your study? Also 63% were treated with ORIF (only 46% were extracapsular fractures), where the time to surgery plays an important role for postoperative complications like avascular necrosis of the femoral head in case of intracapsular fractures.

Reply 2:

The definition of surgical delay adopted in the present study was defined as the specific amount of time elapsed between a patient’s fall and the time of operation. (lines 108-109). Compared with the time duration between patient admission to operation, we considered that fracture-to-operation time reflects diseases seriousness more truly and closing to real world situation. In terms of medical treatment, we have strictly adopted the recommendations published by American Academy of Orthopedic Surgeons (AAOS) in 2015, which stated that “Moderate evidence supports that hip fracture surgery within 48 hours of admission is associated with better outcomes.” (reference: Brox WT, Roberts KC, Taksali S, et al. The American Academy of Orthopaedic Surgeons Evidence-Based Guideline on Management of Hip Fractures in the Elderly. J Bone Joint Surg Am. 2015;97(14):1196-1199. doi:10.2106/JBJS.O.00229) Every participant in the present study received operation within 48 hours after admission.

In short, 294 (78%) patients received hip fracture surgery within 72 hours after injury. (Table 1) Among those 83 patients who received operation over 72 hours after fracture occurred, most of the patient family did not notice or believe that patient had a hip fracture episode. It was not until patient’s physical performance deteriorated did their caregivers send them to hospital for examinations. The top three surgical delay recorded in this study were 3026, 2998, and 933 hours, respectively. Although only 20% of older adults had fracture-to-operation delay > 72 hours, it indicated a potential public health problem of lacking appropriate awareness of hip fracture among older adult’s caregivers in Taiwan.

Question 3:

Discussion: Not all important variables that are mentioned in this study are discussed in the manuscript (e. g. surgical delay, osteoporosis).

Reply 3:

We have added new paragraph in the Discussion section to discuss several important variables (sex, surgical delay, and osteoporosis) which may be associated with older adult’s prognosis following hip fracture surgery. (lines 253-271)

Question 4:

What’s your definition of “length of postoperative recovery”? Length of hospital stay?

Reply 4:

We are sorry for the incorrect wording. We have replaced the word into “length of postoperative follow-up time” throughout the whole manuscript, which would be more appropriate

Round 2

Reviewer 1 Report

Thanks for the corrections made.
Regards.